# $\nu$SAM: Memory-Efficient Sharpness-Aware Minimization via Nuclear Norm Constraints

**Thomas Pethick**                                                      *thomas.pethick@epfl.ch*
*LIONS, IEM, STI, Ecole Polytechnique Fédérale de Lausanne*[*]

**Parameswaran Raman**                                                  *prraman@amazon.com*
*Amazon Web Services*

**Lenon Minorics**                                                      *minorics@amazon.com*
*Amazon Web Services*

**Mingyi Hong**                                                         *mhong@umn.edu*
*University of Minnesota*[†]

**Shoham Sabach**                                                       *ssabach@technion.ac.il*
*Faculty of Data and Decision Sciences, Technion - Israel Institute of Technology*[†]

**Volkan Cevher**                                                       *volkan.cevher@epfl.ch*
*LIONS, IEM, STI, Ecole Polytechnique Fédérale de Lausanne*[†]

**Reviewed on OpenReview:** *https://openreview.net/forum?id=V6ia5hWIMD*

## Abstract

Sharpness-aware minimization (SAM) has been shown to improve the generalization of neural networks. However, the method comes at the expense of storing a perturbation of the model parameters, which can be restrictive when memory bound. We design a variant of SAM, called $\nu$SAM, which obtains a low-rank perturbation by modifying the perturbation constraint. The update almost entirely removes the memory footprint of the perturbation without increasing the computational complexity, thus achieving close to a 1/3 memory saving regarding the parameters when using SGD as the base optimizer. We demonstrate comparable performance of $\nu$SAM with SAM on vision transformers both when training models from scratch and for fine-tuning. Interestingly, $\nu$SAM seems to significantly improve performance for MLP-Mixer architectures across both settings. The results are corroborated theoretically, where we show that SAM with an *arbitrary* norm choice (which includes $\nu$SAM) can converge even with fixed perturbation radius.

## 1 Introduction

Sharpness-aware minimization (SAM) (Foret et al., 2020) has seen rising popularity due to increasing the generalization capability across a wide range of tasks. The method consistently improves classification error (Foret et al., 2020), replaces heavy data augmentation otherwise used in pretraining of vision transformers (ViTs) (Chen et al., 2021), and improves fine-tuning of large language models (LLMs) (Bahri et al., 2021; Zhong et al., 2022). SAM has also been shown to be more robust (Foret et al., 2020), interpretable (Chen et al., 2021) and reproducible (Somepalli et al., 2022).

---

[*]The work of TP was done while interning at Amazon Web Services.
[†]Concurrent positions as an Amazon Scholar and as a faculty at the corresponding institutes. This paper represents the work performed at Amazon.

Table 1: (Left) The number of parameters stored for the perturbation. The $\nu$SAM method uses almost no memory for storing the perturbation in comparison with SAM. (Right) Removing the perturbation entirely would lead to an overall memory saving regarding the model of up to 1/3. *Memory* is provided in multiples of the model size.

| | SAM | $\nu$SAM | Saving |
|---|---|---|---|
| ViT-S/16 | 22M | 75k | 99.66% |
| ViT-B/16 | 87M | 149k | 99.83% |
| ViT-L/16 | 304M | 395k | 99.87% |
| BERT (base) | 110M | 201k | 99.82% |
| BERT (large) | 335M | 479k | 99.86% |

| Method | Memory | Saving |
|---|---|---|
| SGD | 2 | $^1/_3$ |
| SGD momentum | 3 | $^1/_4$ |
| Adam(W) | 4 | $^1/_5$ |
| Adam(W) bf16 momentum | 3 | $^1/_4$ |
| Adafactor | $\sim 2$ | $^1/_3$ |

The update rule of SAM proceeds by finding a (norm constrained) perturbation vector $\varepsilon \in \mathbb{R}^d$ and subsequently updating the weights $x \in \mathbb{R}^d$ using a gradient $\nabla f$ computed at the perturbed set of weights,

$$x \leftarrow x - \gamma \nabla f(x + \varepsilon). \tag{1}$$

Despite its popularity, one subtle problem is the increased memory footprint of SAM as compared to conventional first-order methods such as stochastic gradient descent (SGD). Specifically, at any given time the SAM method needs to store: the weights, a gradient and *additionally* the perturbation. This is 50% more memory demanding in terms of the weights than SGD, which only needs to store the weights and a gradient.

This naturally raises the following research questions:

> *Is it possible to obtain a SAM formulation that induces a memory-efficient algorithm without introducing additional computational overhead?*

In this work, we answer the above question in the affirmative. Concretely we make the following contributions:

- By revisiting the original SAM formulation we notice that the structure of the network is lost in the computation of the perturbation $\varepsilon$ since the parameters are treated as a vector. This motivates us to replace the original vector $\ell_2$-norm constraint with a particular matrix norm constraint (specifically the nuclear norm) that allows us to obtain a layerwise low-rank $\varepsilon$. The modification leads to substantial memory-savings regarding the perturbation, saving more than 99.8% of the memory required by the perturbation in the original SAM method on ViT-B/16 (see Table 1).

- We extensively evaluate our method $\nu$SAM on vision transformers (ViTs) and MLP-Mixer models when both fine-tuning and training from scratch, and additionally fine-tune BERT on a set of language tasks. We find that $\nu$SAM consistently outperforms the baseline AdamW in all cases and achieves comparable performance with SAM. Surprisingly, this is the case even when the low-rank decomposition of the perturbation $\varepsilon$ is only coarsely approximated through a single power iteration, which avoid adding *any* wall-clock time as compared with SAM. Interestingly, we find that $\nu$SAM enjoys a substantial improvement over SAM for MLP-Mixer models on both fine-tuning and when training from scratch.

- We provide a strong baseline for ViTs on relatively small datasets like CIFAR, which might be of independent interest, increasing the baseline in (Mueller et al., 2023) by more than 3pp (see Sections 5.1 and 5.2.1). We interestingly find that, in this setting, both $\nu$SAM and SAM *substantially* improve if the perturbation is not activated until after several epochs. The delay raises the percentage points improvement over AdamW by a striking factor of $\sim 4$. It appears that the perturbation can be taken larger if delayed, partially explaining the benefit.

- Theoretically, we find that SAM-type methods with *arbitrary* norm choice for the perturbation (which includes $\nu$SAM) converges even with fixed perturbation radius $\rho$, although only for a restrictive class of certain convex quadratics. This is in stark contrast with a SAM variant using

(normalized) random perturbation, which is not guaranteed to converge, as we demonstrate. This observation possibly sheds some light on the practically observed advantage of using gradient information in the perturbation.

**Limitations** Our work focuses on transformers and MLP-Mixer models, since our memory-efficient reformulation of SAM heavily relies on the presence of matrix structures in the network.

## 2 Related work

**Memory-efficient SAM** There has been other works trying to lower the memory consumption of SAM. FSAM (Zhong et al., 2022) and SSAM (Mi et al., 2022) use the Fisher information to identify a subset of the parameters to perturb. In practice SSAM can save 50%, and FSAM roughly 90%, of the perturbation (see Zhong et al. (2022, Sec. 6.1)) whereas $\nu$SAM saves $\sim 99.8\%$ (see Table 1). Very recently, a similar memory saving as $\nu$SAM was made possible with SAM-ON, which only perturbs the normalization layers (Mueller et al., 2023). Very surprisingly, this minor modification leads to improvements over SAM in many settings. We compare in detail in the experimental section where we find that our method has an advantage for specifically MLP-Mixer architectures and for fine-tuning tasks across architectures.

**Convergence of SAM** The analysis in GSAM (Zhuang et al., 2022), SSAM (Mi et al., 2022) and Andriushchenko & Flammarion (2022, Thm. 2) takes decreasing perturbation radius $\rho_k$. Almost sure convergence of SAM was shown in Nam et al. (2023) using a similar construction. The decreasing $\rho_k$ was avoided in (Andriushchenko & Flammarion, 2022, Thm. 6) by ignoring the normalization in SAM. Very recently, SAM with fixed perturbation radius $\rho$ was studied in Si & Yun (2023) with convergence guarantees for (strongly)-convex objectives, while providing negative results for stochastic and nonconvex cases.

**Power iteration** An important method for eigenvalue and singular value computation is the power iteration method (Mises & Pollaczek-Geiringer, 1929) (see e.g. Golub & Van der Vorst (2000)). The method has been extensively used in the machine learning community. It is, for instance, the backbone of the PageRank algorithm (Page et al., 1999), and used for computing the spectral norm in training of generative adversarial networks (Miyato et al., 2018).

**Low-rank approximations** There have been several work exploiting low rank structures in machine learning application: Vogels et al. (2019) uses a low rank approximation to save on communication costs in distributed settings, recently low-rank fine-tuning was popularized by LoRA (Hu et al., 2021), and other work explores the effect of initialization on low-rank pretraining (Kamalakara et al., 2022).

## 3 Algorithmic derivation

The starting point of SAM is the following saddle point problem for a given loss $f : \mathcal{X} \to \mathbb{R}$

$$\min_{x \in \mathcal{X}} \max_{\varepsilon \in \mathcal{X} : \|\varepsilon\| \leq \rho} f(x + \varepsilon), \tag{2}$$

where $\rho \in [0, \infty)$ is the perturbation radius and $\|\cdot\|$ is some norm to be defined. To obtain a computationally efficient method, SAM (Foret et al., 2020) linearizes the maximization problem with a first-order Taylor expansion as follows

$$\varepsilon^\star \in \underset{\varepsilon \in \mathcal{X} : \|\varepsilon\| \leq \rho}{\text{Arg max}} f(x + \varepsilon) \approx \underset{\varepsilon \in \mathcal{X} : \|\varepsilon\| \leq \rho}{\text{Arg max}} \{f(x) + \langle \varepsilon, \nabla f(x) \rangle\} = \underset{\varepsilon \in \mathcal{X} : \|\varepsilon\| \leq \rho}{\text{Arg max}} \langle \varepsilon, \nabla f(x) \rangle . \tag{3}$$

For the particular choice of $\ell_2$-norm constraints, the argmax has a closed form solution, which leads to the following update rule:

$$\boxed{\begin{aligned} \varepsilon^k &= \tfrac{\rho}{\|\nabla f(x^k)\|_2} \nabla f(x^k), \\ x^{k+1} &= x^k - \gamma \nabla f(x^k + \varepsilon^k). \end{aligned}} \tag{SAM}$$

---

**Algorithm 1** Nuclear norm based sharpness-aware minimization ($\nu$SAM)

---

**Require:** Parameter initialization $x^{-1} = (M_1^{-1}, \ldots, M_l^{-1})$, SVD initialization $u_i^{-1} \in \mathbb{R}^{n_i} \; \forall \, i \in [l]$
**Repeat** for $k = 0, 1, \ldots, K - 1$
  1: Draw a sample $\xi_k \sim \mathcal{P}$
  2: Compute the adversarial perturbation $\widetilde{x}^{k+1} = (\widetilde{M}_i^{k+1})_{i \in [l]}$:

$$
\begin{cases}
u_i^k, v_i^k & = \mathrm{SVD}_{\mathrm{top1}}(\nabla_{M_i} f(x^k, \xi_k), u_i^{k-1}) \\
\widetilde{M}_i^{k+1} & = M_i^k + \rho u_i^k (v_i^k)^\top
\end{cases} \quad \forall i \in [l]
$$

  3: Update the weights $x^{k+1} = (M_i^{k+1})_{i \in [l]}$1[1]:

$$
M_i^{k+1} = \widetilde{M}_i^{k+1} - \rho u_i^k (v_i^k)^\top - \gamma \nabla_{M_i} f(\widetilde{x}^{k+1}, \xi_k) \quad \forall i \in [l]
$$

**Return** $x^K := (M_1^K, \ldots, M_l^K)$
[1]For simplicity the update is only for matrix shaped parameters. Update non-matrix parameters using gradient descent.

---

**Algorithm 2** Top singular value decomposition ($\mathrm{SVD}_{\mathrm{top1}}$)

---

**Require:** Matrix $A \in \mathbb{R}^{n \times m}$, initialization $u_{-1} \in \mathbb{R}^n$, $\tau = 10^{-12} \in (0, \infty)$
**Repeat** for $t = 0, 1, \ldots, T - 1$
  1: $v^{t+1} = \frac{A^\top u^t}{\|A^\top u^t\|_2 + \tau}$
  2: $u^{t+1} = \frac{A v^{t+1}}{\|A v^{t+1}\|_2 + \tau}$
**Return** $u^T, v^T$

---

One potential limitation of SAM is the requirement to store one entire additional set of model parameters. To economize on memory we revisit (3) and notice that the RHS can be interpreted as a linear minimization oracle (LMO), i.e.,

$$
\mathrm{lmo}_{\mathcal{Z}}(s) := \operatorname*{Arg\,min}_{z \in \mathcal{Z}} \langle z, s \rangle. \tag{4}
$$

To be precise, (3) can be written as

$$
\operatorname*{Arg\,max}_{\varepsilon \in \mathcal{X}: \|\varepsilon\| \le \rho} \langle \varepsilon, \nabla f(x) \rangle = \mathrm{lmo}_{\mathcal{E}}(-\nabla f(x)), \tag{5}
$$

where $\mathcal{E} = \{\, \varepsilon \in \mathcal{X} \mid \|\varepsilon\| \le \rho \,\}$. It is well-known that for certain choices of norms, such as the nuclear norm, the LMO leads to a sparse solution, which we will exploit in what follows.

Abstractly, with a stepsize $\gamma > 0$, our update takes the following form

$$
\begin{aligned}
\varepsilon^k &= \mathrm{lmo}_{\mathcal{E}}(-\nabla f(x^k)), \\
x^{k+1} &= x^k - \gamma \nabla f(x^k + \varepsilon^k).
\end{aligned} \tag{6}
$$

Instead of flattening the model parameters, we will consider the case where our model is parametrized by a matrix in $\mathcal{X} = \mathbb{R}^{n \times m}$ and we choose the nuclear norm constraint $\mathcal{E}_* = \{\, \varepsilon \in \mathcal{X} \mid \|\varepsilon\|_* \le \rho \,\}$. In that case update (6) reduces to only computing

$$
\varepsilon^k = \rho u^k (v^k)^\top, \tag{7a}
$$
$$
x^{k+1} = x^k - \gamma \nabla f(x^k + \varepsilon^k). \tag{7b}
$$

where $u^k$ and $v^k$ are the singular vectors associated with the top singular value of $-\nabla f(x^k)$ (see Section 3.1 for derivations).

We crucially rearranged the update to avoid ever storing $\varepsilon^k$ and $x^k$ simultaneously:

$$\widetilde{x}^k = x^k + \rho u^k (v^k)^\top, \tag{8a}$$
$$x^{k+1} = \widetilde{x}^k - \rho u^k (v^k)^\top - \gamma \nabla f(\widetilde{x}^k). \tag{8b}$$

In other words, $x^k$ can be completely discarded after step 2 in $\nu$SAM (Algorithm 1), which is critical to achieving the memory saving. In (8a), we only need to store one matrix $x^k$ and two vectors $u^k$ and $v^k$ by performing the update in-place. Similarly, in (8b), it suffice to store the matrices $x^k$ and $\nabla f(\tilde{x}^k)$ and two vectors $u^k$ and $v^k$. In contrast, (SAM) requires storing three matrices simultaneously.

The top singular vectors can efficiently be computed using (warmstarted) power iterations (Algorithm 2). We simply ignore the update for all other parameters including the vector-valued bias terms in the MLP layers. The resulting algorithm is described in $\nu$SAM (pronounced "newSAM") where the weight parameterization consists of $l$ weight matrices $M_i \in \mathbb{R}^{n_i \times m_i}$, i.e., $x = (M_1, \ldots, M_l)$ and the update uses stochastic gradients.

As seen in $\nu$SAM (Algorithm 1) the LMO is applied *independently* to each layer matrix, which is beneficial for distributed settings (see Section 3.2). We note that the resulting update can still be interpreted as a norm-constrained LMO on the joint parameters $x$, namely as the max-norm over nuclear norms,

$$\|x\|_{\mathcal{X}} := \max\{\|M_1\|_*, \|M_2\|_*, \ldots, \|M_\ell\|_*\}.$$

Consequently, the convergence in Theorem 4.3 applies. We also develop and compare with a normalized variant of $\nu$SAM in Appendix C.1, where layer updates are no longer independent.

The memory savings are large if the embedding dimensions (i.e., layer width) are large (c.f. Table 1). The low-rank perturbation allows $\nu$SAM to reduce the memory consumption associated with storing the perturbation by almost an order $10^3$, e.g. the 304M parameter ViT-L/16 model only requires 395k parameters for the perturbation.

### 3.1 The low-rank update

To arrive at (8a) we seek a closed form solution to the linear minimization oracle (LMO) under a nuclear norm constraint,

$$\mathrm{lmo}\,(W) := \underset{X:\|X\|_* \leq \rho}{\arg\min} \; \langle W, X \rangle. \tag{9}$$

We include the closed form solution below for completeness (see e.g. Jaggi (2013) for a similar result).

**Lemma 3.1.** *The LMO under nuclear constraint in* (9) *can be efficiently computed as*

$$\mathrm{lmo}\,(W) = -\rho u v^\top,$$

*where $u$ and $v$ are the left and right singular vectors associated with the top singular value of $W$.*

Lemma 3.1 yields the update rule in (7), which, when approximated with power iterations, leads to Algorithm 1.

### 3.2 Implementation

In practice we avoid perturbing the bias term and the (single) convolutional layer in ViTs if not otherwise specified.

See Algorithm 3 in Appendix C for pseudo-code for a distributed setting with distributed data parallel (DDP). We note that $\nu$SAM is also compatible with fully sharded data parallel (FSDP) as long as the neural network structure is preserved when sharded, such that the matrix structure is not lost. Compatibility with FSDP opens up the possibility to apply $\nu$SAM to models that do not fit on a single GPU. As of writing, the FairScale implementation (FairScale authors, 2021) allows preserving the structure whereas the PyTorch implementation flattens the network parameters (v2.0.0). One favorable property of $\nu$SAM under FSDP is that no global normalization needs to be computed and distributed between the layer computations, which is otherwise the case for SAM.

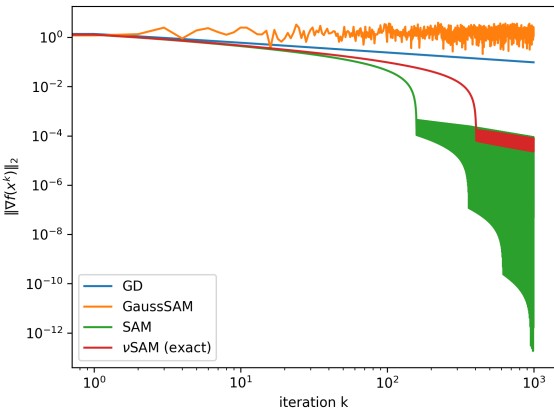

Figure 1: Demonstration of Theorem 4.3. The direction of the perturbation is important: Both $\nu$SAM and SAM with a decreasing stepsize $\gamma_k = 1/k$ enjoys a $\mathcal{O}(1/k^2)$ rate on $\min_{i \in [k-1]} \|\nabla f(x^i)\|^2$, while perturbing with normalized Gaussian noise (GaussSAM) exhibits nonconvergence. Gradient descent (GD) with the same stepsize only converges as $\mathcal{O}(1/k)$.

The $\nu$SAM method almost entirely removes the memory overhead of the perturbation. It is important, in conjunction, to also reduce the memory footprint of the optimizer states. This work uses the popular Adam (Kingma, 2014) and AdamW (Loshchilov & Hutter, 2017) as the base optimizer, but Adam(W) can be readily replaced by a memory efficient variant, such as Adafactor (Shazeer & Stern, 2018), which is commonly used in the vision transformer literature (see e.g., (Zhai et al., 2022, sec. 3.4) and Beyer et al. (2022)). For fine-tuning specifically, we note that $\nu$SAM is also compatible with using low-rank updates for the *minimizer* as in e.g., LoRA (Hu et al., 2021). We provide an overview of methods in Table 1, which shows that the relative memory saving is higher when the base optimizer has a small memory footprint.

## 4    Convergence analysis

The gradient direction in the perturbation appearing through the LMO of both SAM and $\nu$SAM turns out to be important for convergence under fixed perturbation radius $\rho$. If a random direction is used instead, convergence cannot be guaranteed as illustrated in Figure 1. We will now make this claim precise.

In this section, $\|\cdot\|_*$ will more generally denote the dual norm of some norm $\|\cdot\|$, instead of specifically referring to the nuclear norm. For the matrix case $\|\cdot\|_2$ refers to the entry-wise 2-norm (i.e., the Frobenius norm).

Abstractly our update takes the following form:

$$
\begin{aligned}
\widetilde{x}^k &= x^k + \rho_k \varepsilon^k, \\
x^{k+1} &= x^k - \gamma_k \nabla f(\widetilde{x}^k),
\end{aligned}
\tag{10}
$$

where the particular choice of $\varepsilon^k \in \mathcal{X}$ is to be defined.

Interestingly, we can show convergence *without* decreasing $\rho$ for certain convex quadratics, if we restrict the $\varepsilon$-perturbation to the LMO over an (arbitrary) norm-ball.

We make the following assumptions and define $f^\star := \inf_{x \in \mathcal{X}} f(x)$.

**Assumption 4.1.** *The function $f : \mathcal{X} \to \mathbb{R}$ is strongly-convex with parameter $\mu > 0$, i.e.,*

$$
\langle \nabla f(x) - \nabla f(x'), x - x' \rangle \geq \mu \|x - x'\|_2^2 \quad \forall x, x' \in \mathcal{X}.
$$

**Assumption 4.2.** *The gradient* $\nabla f : \mathcal{X} \to \mathcal{X}$ *is L-Lipschitz with* $L > 0$, *i.e.,*

$$\|\nabla f(x) - \nabla f(x')\|_2 \le L\|x - x'\|_2 \quad \forall x, x' \in \mathcal{X}.$$

**Theorem 4.3.** *Suppose Assumptions 4.1 and 4.2 hold with* $L = \mu$. *Then,* (10) *with the (arbitrary) norm perturbation* $\varepsilon^k \in \text{Arg}\max_{\varepsilon:\|\varepsilon\|_* \le 1} \langle \nabla f(x^k), \varepsilon \rangle$ *satisfies the following descent inequality with* $\rho \ge 0$ *and* $\gamma_k > 0$,

$$f(x^{k+1}) - f^\star \le (1 - \mu\gamma_k)(f(x^k) - f^\star) - \gamma_k(1 - \gamma_k L)L\rho\|\nabla f(x^k)\| + \frac{\gamma_k^2 L^3 \rho^2 \xi_*^2}{2}.$$

*Furthermore, for* $\gamma_k = \gamma = \min\{\frac{1}{\mu K}\log(\alpha), \frac{1}{2L}\}$ *with* $\alpha = \frac{\Delta_0 2\mu^3 K^2}{L^4 \rho^2 \xi_*^4 \xi^2}$, *it follows that*

$$\min_{k \in [K]} f(x^k) - f^\star = \widetilde{\mathcal{O}}\left(\exp(-\tfrac{K}{2})\Delta_0 + \tfrac{C\rho^2}{K^2}\right)$$

*with* $\Delta_0 = f(x^0) - f^\star$, $C = L\xi_*^4 \xi^2$, $\xi = \max_{x \in \mathcal{X}} \|x\|_2/\|x\|$ *and* $\xi_* = \max_{x \in \mathcal{X}} \|x\|_2/\|x\|_*$.

*Remark* 4.4. Theorem 4.3 holds for *any* norm-ball constrained perturbation including both SAM and $\nu$SAM. In this sense, Theorem 4.3 can be seen as a generalization of the recent result of Si & Yun (2023). Interestingly, the fact that the proof uses the particular direction of the perturbation appears to not only be an artifact of the proof technique. In particular, if the perturbation is instead replaced with normalized Gaussian noise, the scheme does not converge (cf. Figure 1).

The assumptions of Theorem 4.3 are rather restrictive, but they are sufficient for showing the importance of including the gradient direction in the perturbation. Theorem 4.3 applies to quadratic minimization problem of the form $f(x) = \frac{L}{2}\|x - b\|^2 + c$ for some $b \in \mathcal{X}$ and $c \in \mathbb{R}$, which we demonstrate it on in Figure 1. The parameter $\rho$ is kept fixed and the stepsize is taken as $\gamma_k = \mathcal{O}(1/k)$ as suggested by the theory.

The particular direction of the perturbation turns out to be important not only for the proof. If normalized Gaussian noise is used (i.e., $\varepsilon^k = e^k/\|e^k\|_2$ with $e^k \sim \mathcal{N}(0, I)$ denoted as GaussSAM) we observe that the iterates only converges within a neighborhood of the solution as also suggested by theory (cf. e.g. (Li et al., 2024, Thm. 4)). This negative result is particularly interesting in the light of the generalization bound of SAM in Foret et al. (2020, Thm. 1), whose statement would also hold for GaussSAM. It is therefore not obvious from the generalization bound alone that the gradient direction in both $\nu$SAM and SAM is preferred over GaussSAM. The optimization perspective in Theorem 4.3 provides a possible explanation, by showing that only the former enjoys convergence guarantees. In addition, we observe the fast $\mathcal{O}(1/k^2)$ rate for $\nu$SAM and SAM, whereas the (unperturbed) gradient descent (GD) with the same stepsize has the slower $\mathcal{O}(1/k)$ rate.

If the perturbation radius can be taken decreasing it is possible to show convergence *without* convexity assumptions, as long as the gradients are bounded. No structural assumptions are needed on the perturbation $\varepsilon^k$ (apart from being bounded) and the result thus also applies to $\nu$SAM.

**Assumption 4.5.** *The gradient* $\nabla f : \mathcal{X} \to \mathcal{X}$ *is bounded, i.e.,* $\|\nabla f(x)\|_2 \le G \ \forall x \in \mathcal{X}$.

**Theorem 4.6.** *Suppose Assumptions 4.2 and 4.5 hold and* $\|\varepsilon^k\| \le \xi$ *for all* $k \in [K]$. *Then* (10) *satisfies the following convergence guarantee for* $\rho_k \ge 0$ *and* $\gamma_k \in (0, 2/L)$,

$$\min_{k=0,\ldots,K-1} \|\nabla f(x^k)\|_2^2 \le \frac{\Delta_0 + C_1 \sum_{k=0}^{K-1} \gamma_k^2 \rho_k^2 + C_2 \sum_{k=0}^{K-1} \gamma_k \rho_k}{\sum_{k=0}^{K-1} \gamma_k(1 - \gamma_k L/2)}$$

*with* $\Delta_0 = f(x^0) - f^\star$, $C_1 = \frac{1}{2}L^3\xi^2$ *and* $C_2 = LG\xi$.

*In particular, for constant stepsize* $\gamma_k = \gamma = 1/L\sqrt{K}$ *and* $\rho = \frac{1}{\sqrt{K}}$, *the following rate is obtained*

$$\min_{k=0,\ldots,K-1} \|\nabla f(x^k)\|_2^2 = \mathcal{O}\left(\tfrac{\Delta_0}{\sqrt{K}} + \tfrac{C_2}{\sqrt{K}} + \tfrac{C_1}{LK^{3/2}}\right)$$

*Remark* 4.7. To get convergence one needs to take $\rho_k$ decreasing. Specifically, it suffice to take $\sum_{k=0}^{\infty} \gamma_k \rho_k < \infty$ and $\sum_{k=0}^{\infty} \gamma_k = \infty$. We note that decreasing $\rho$ is used for SOTA experimental SAM results as in Zhuang et al. (2022). Theorem 4.6 also implies a fallback guarantee for when $\rho$ is taken constant, since the theorem in that case ensures convergence to a $\rho$-dependent neighborhood.

Table 2: Training from scratch on CIFAR10/100 where $\nu$SAM shows strong performance on especially Mixer models where test accuracy is increased over SAM by between 0.28pp and 0.55pp, while memory usage associated with the perturbation is reduced by almost 3 orders of magnitude.

| Dataset | Method Model | AdamW | SAM | SAM-ON | $\nu$SAM |
|---|---|---|---|---|---|
| CIFAR10 | Mixer-B/4 | 91.00 | 91.70 | 91.73 (+0.03) | **92.23** (+0.53) |
| | Mixer-S/4 | 91.08 | 91.96 | 92.00 (+0.04) | **92.24** (+0.28) |
| | ViT-B/4 | 93.45 | 94.62 | **94.90** (+0.28) | 94.44 (-0.18) |
| | ViT-S/4 | 93.37 | 94.21 | **94.65** (+0.44) | 94.39 (+0.18) |
| CIFAR100 | Mixer-B/4 | 66.90 | 69.54 | 68.25 (-1.29) | **70.09** (+0.55) |
| | Mixer-S/4 | 68.03 | 70.35 | 69.35 (-1.00) | **70.79** (+0.44) |
| | ViT-B/4 | 68.25 | 70.26 | **71.25** (+0.99) | 70.15 (-0.11) |
| | ViT-S/4 | 68.17 | 71.71 | **72.70** (+0.99) | 70.60 (-1.11) |

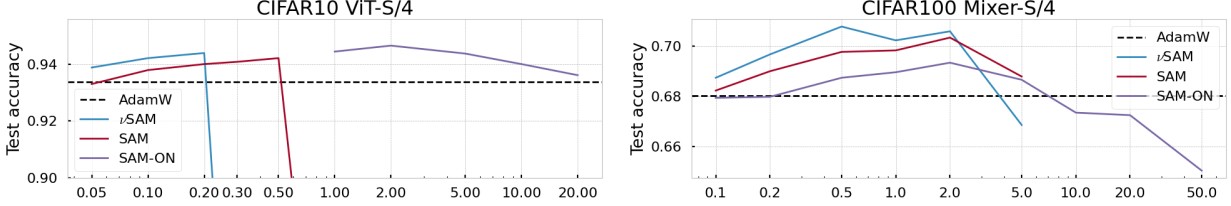

Figure 2: Sweep over $\rho$ for pretraining results in Table 2. See Figure 4 in Appendix D for remaining configurations.

## 5 Experiments

We evaluate $\nu$SAM both when training from scratch and for fine-tuning against several baselines, namely Adam(W) (Kingma, 2014; Loshchilov & Hutter, 2017), SAM (Foret et al., 2020) and SAM-ON (Mueller et al., 2023). The relative (colored) number in the tables captures the difference in performance from SAM.

Note that we use a single power iteration ($T = 1$ in Algorithm 2) for the approximation in $\nu$SAM if not otherwise stated and only the weight *matrices* are perturbed. Specifically, we simply ignore the (single) convolutional layer in the ViT and the bias terms when applying $\nu$SAM. This leads to a method that has a wall-clock time matching that of SAM (c.f. Table 14 in Appendix D.1). We experiment with using more power iterations in Table 10 of Appendix D, but conclude that $T = 1$ performs sufficiently well, while remaining computationally cheap.

### 5.1 Training from scratch

We train multiple sizes of ViTs and MLP-Mixer on CIFAR10/100 from scratch. We specifically take the original architecture configuration of ViTs and MLP-Mixer models in Dosovitskiy et al. (2020); Tolstikhin et al. (2021) and downsize the patch size to 4 to fit the smaller image size of $32 \times 32$. See Table 6 in Appendix D for details.

**Baseline & hyperparameters** Since there is a lack of good hyperparameter defaults for these architectures on small datasets, we first find a good configuration for the base optimizer AdamW on CIFAR10. We provide a substantially better baseline (AdamW) than e.g. Mueller et al. (2023, Table 3) (93.37% instead of 90.34% on ViT-S/4). The baseline is comparable with SOTA ViTs for small datasets (Gani et al., 2022; Lee et al., 2021; Liu et al., 2021) without optimizing the ViT structure and without using CutMix, MixUp, repeated augment, stochastic depth and random erase. The final hyperparameters can be found in Table 7 of Appendix D. We use standard augmentations (random cropping and flipping) and AutoAugment as in Gani et al. (2022) and a cosine learning rate schedule with linear warmup.

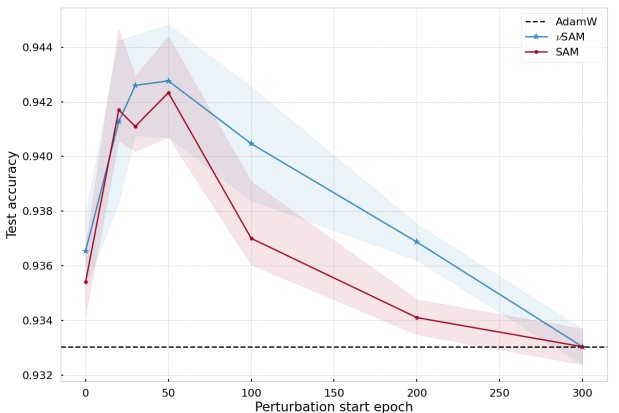

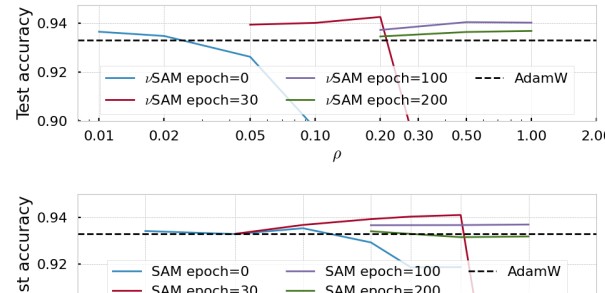

(a) Performance of the best $\rho$ for a given perturbation delay.

(b) The optimal $\rho$ for $\nu$SAM increases with the perturbation start epoch. Both SAM and $\nu$SAM suffer if activated immidately.

Figure 3: Both SAM and $\nu$SAM benefit from delaying perturbing until after the learning rate warmup phase.

Table 3: Pretraining without AutoAugment. The gap in comparison with AdamW is larger and $\nu$SAM provides a substantial improvement over SAM, while significantly reducing memory usage.

| Model | Method
Dataset | AdamW | SAM | SAM-ON | $\nu$SAM |
|---|---|---|---|---|---|
| Mixer-S/4 | CIFAR10 | 85.79 | 87.92 | 86.18 (-1.74) | **88.55** (+0.63) |
| | CIFAR100 | 59.09 | 60.73 | 60.52 (-0.21) | **61.09** (+0.36) |
| ViT-S/4 | CIFAR10 | 88.49 | 89.90 | **90.74** (+0.84) | 90.42 (+0.52) |
| | CIFAR100 | 61.22 | 65.11 | **66.01** (+0.90) | 65.29 (+0.18) |

**Comparison of SAM variants** In order to fairly compare variants of SAM, we sweep over the hyperparameter $\rho$ and pick the best one for each combination of model type, dataset and method (c.f. Figure 4 in Appendix D). We discover that $\rho$ can be significantly larger for both SAM and $\nu$SAM if the perturbation is only started *after* the learning rate warmup period. Delaying the perturbation increases the test accuracy for both methods, so we use this as the default throughout our experiments. We further investigate this curious phenomenon in Section 5.1.1.

**Results** The test accuracies of the best iterate are shown in Table 2. The $\nu$SAM method yields a large improvement on the MLP-Mixer architecture, which is maybe not surprising since there are no convolutional layers as in ViTs. What is maybe surprising, is that $\nu$SAM performs substantially better than *all* the baselines, including SAM, on MLP-Mixer across both datasets. We additionally provide experiments without AutoAugment on ViT-S/4 in Table 3 where $\nu$SAM is found to consistently outperform SAM.

### 5.1.1 Delaying the perturbation

We notice that delaying the perturbation in SAM and $\nu$SAM until after the learning rate warmup phase greatly improves the test accuracy in our training setup. This is in contrast with earlier observations (Agarwala & Dauphin, 2023). To investigate the effect of the delay we sweep over multiple radii, $\rho$, for each delay to ensure optimality. The results can be found in Figure 3 which uses a horizon of 300 epochs on CIFAR10 and a ViT-S/4. We compute a mean and standard deviation over 3 independent runs for each configuration.

Interestingly, the best $\rho$ for $\nu$SAM increases as the delay is increased. It appears that perturbing in the early training phase is problematic for both $\nu$SAM and SAM. Therefore, we delay perturbation for both method until after 10% of the epochs (after the learning rate warmup). Curiously, SAM-ON (Mueller et al., 2023) on the other hand does not seem to benefit from a similar delay (c.f. Table 9 of Appendix D).

Table 4: Fine-tuning on CIFAR10/100 where $\nu$SAM consistently outperforms SAM-ON and provides a significant performance boost over SAM for the MLP-Mixer model on CIFAR10 (0.21pp) and CIFAR100 (0.68pp).

| Dataset | Method Model | AdamW | SAM | SAM-ON | $\nu$SAM |
|---|---|---|---|---|---|
| CIFAR10 | Mixer-B/16 | 97.16 ± 0.16 | 97.52 ± 0.08 | 97.58 (+0.06) ± 0.10 | **97.72** (+0.21) ± 0.05 |
| | ViT-B/16 | 98.36 ± 0.19 | 98.87 ± 0.05 | 98.74 (-0.13) ± 0.10 | **98.89** (+0.02) ± 0.03 |
| | ViT-S/16 | 98.38 ± 0.02 | **98.85** ± 0.00 | 98.73 (-0.12) ± 0.01 | 98.81 (-0.04) ± 0.02 |
| CIFAR100 | Mixer-B/16 | 85.87 ± 0.05 | 86.20 ± 0.19 | 86.39 (+0.19) ± 0.17 | **86.88** (+0.68) ± 0.25 |
| | ViT-B/16 | 90.54 ± 0.03 | **92.12** ± 0.13 | 91.47 (-0.65) ± 0.05 | 91.89 (-0.23) ± 0.13 |
| | ViT-S/16 | 91.10 ± 0.12 | **91.74** ± 0.23 | 91.37 (-0.37) ± 0.13 | 91.60 (-0.14) ± 0.05 |

Table 5: Fine-tuning of BERT-base (uncased) on GLUE.

| | Adam | SAM | SAM-ON | $\nu$SAM | Length |
|---|---|---|---|---|---|
| CoLA | 57.31 ± 1.58 | 57.02 ± 1.41 | **57.84** (+0.82) ± 0.89 | 57.38 (+0.37) ± 1.12 | 8.5k |
| MNLI-m | 83.82 ± 0.04 | **84.23** ± 0.11 | 84.08 (-0.14) ± 0.27 | 84.03 (-0.20) ± 0.23 | 393k |
| MNLI-mm | 84.02 ± 0.29 | **84.37** ± 0.16 | 83.98 (-0.39) ± 0.29 | 84.16 (-0.21) ± 0.29 | 393k |
| MRPC | 85.11 ± 0.76 | 86.13 ± 1.12 | 85.64 (-0.49) ± 1.17 | **86.42** (+0.29) ± 0.96 | 3.7k |
| MRPC (F1) | 89.56 ± 0.58 | 90.31 ± 0.77 | 89.95 (-0.36) ± 0.74 | **90.46** (+0.15) ± 0.69 | 3.7k |
| QNLI | 90.20 ± 0.82 | 90.87 ± 0.47 | **90.90** (+0.03) ± 0.48 | 90.11 (-0.76) ± 0.27 | 105k |
| QQP | 91.03 ± 0.11 | **91.42** ± 0.23 | 91.29 (-0.13) ± 0.05 | 91.29 (-0.13) ± 0.09 | 364k |
| QQP (F1) | 87.94 ± 0.11 | **88.47** ± 0.33 | 88.30 (-0.17) ± 0.08 | 88.32 (-0.14) ± 0.11 | 364k |
| RTE | 58.12 ± 0.98 | **61.01** ± 2.35 | 60.58 (-0.43) ± 2.05 | 59.93 (-1.08) ± 0.57 | 2.5k |
| SST-2 | 92.63 ± 0.83 | **92.78** ± 0.69 | 92.27 (-0.50) ± 0.52 | 92.68 (-0.09) ± 0.55 | 67k |
| STS-B (Pearson) | 87.34 ± 0.18 | **87.43** ± 0.20 | 87.41 (-0.02) ± 0.32 | 87.43 (-0.01) ± 0.43 | 7k |
| STS-B (Spearman) | 87.06 ± 0.21 | 87.18 ± 0.19 | 87.16 (-0.02) ± 0.31 | **87.26** (+0.08) ± 0.37 | 7k |
| Avg | 81.22 ± 0.35 | **81.83** ± 0.17 | 81.70 (-0.13) ± 0.37 | 81.61 (-0.22) ± 0.13 | - |

## 5.2 Fine-tuning

We fine-tune on both vision tasks and language tasks across multiple datasets.

### 5.2.1 Vision task

**Setup** We take ViT and MLP-Mixer architectures pretrained on ImageNet and fine-tune the models on CIFAR10/100. Since there is no pretrained Mixer-S/16 release from Tolstikhin et al. (2021); Steiner (2022) we restrict the experiments concerning MLP-Mixer architectures to Mixer-B/16. We optimize the baseline AdamW, whose final hyperparameters can be found in Table 11 of Appendix D. We sweep over the perturbation radius $\rho$ for all perturbation-based methods for a fair comparison (see Figure 5 in Appendix D for $\rho$ sweep). Each configuration is run 3 times to computes a mean and standard deviation.

Our baseline performs substantially better than similar experiments in Zhuang et al. (2022, Table 3). The improvement upon the baseline is primarily due to the use of a smaller learning rate and no weight decay. The configuration additionally allows us to run for only 10 epochs (20 times less iterations) while still achieving better performance.

**Results** The results are shown in Table 4. Maybe surprisingly, $\nu$SAM seems to especially provide an improvement for the larger `base` model size. Similar to training from scratch in Section 5.1 we also see strong performance on the MLP-Mixer architecture. Even if AdamW is optimized further and given a larger computational budget it does not close the performance gap (see Table 13 in Appendix D).

### 5.2.2 Language task

We fine-tune a pretrained BERT-base (uncased) (Devlin et al., 2018) on the GLUE benchmark (Wang et al., 2018) following the setup and hyperparameters in Geiping & Goldstein (2022). We use a 10% perturbation delay for SAM and $\nu$SAM (c.f. Section 5.1.1) and sweep over $\rho \in \{0.01, 0.02, 0.05\}$. The best configuration is picked and run 5 times to provide a mean and standard deviation. Hyperparameters can be found in Table 12 of Appendix D. The results are shown in Table 5 where we find that all SAM variants improve upon the Adam baseline. The $\nu$SAM method appears to exhibit the smallest variance across all the methods on average.

## 6 Conclusion

We developed $\nu$SAM, a sharpness-aware minimization algorithm that almost entirely removes the additional memory otherwise required for storing the perturbation. The method uses a low-rank approximation that only needs to be very coarsely approximated in practice, resulting in no runtime overhead compared with the original SAM method. We observe strong performance on particularly MLP-Mixer models across both fine-tuning and when training from scratch. We additionally find that both $\nu$SAM and SAM benefit from delaying activating perturbation in certain settings. Interesting future work involves better understanding the effect of the architecture and the benefit of perturbation delay as well as exploring the use of other norm choices.

## 7 Acknowledgments

S. Sabach, M. Hong, and V. Cevher hold concurrent appointments as an Amazon Scholar and as a faculty at Technion, University of Minnesota, and EPFL, respectively.

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

# Appendix

## Table of Contents

## A  Proofs for Section 3 (Algorithmic derivation)

**Lemma 3.1.** *The LMO under nuclear constraint in* (9) *can be efficiently computed as*

$$\mathrm{lmo}\,(W) = -\rho u v^\top,$$

*where $u$ and $v$ are the left and right singular vectors associated with the top singular value of $W$.*

*Proof.* We are asked to find $X$ satisfying $\min_{\|X\|_* \leq \rho} \langle W, X \rangle$, which relates to the definition of the dual norm,

$$\frac{1}{\rho} \min_{\|X\|_* \leq \rho} \langle W, X \rangle = \max_{\|\bar{X}\|_* \leq 1} \langle W, \bar{X} \rangle =: \|W\| \quad \text{where} \quad \bar{X} := -\frac{1}{\rho} X. \tag{11}$$

where $\|\cdot\|$ is the spectral norm. We can rewrite (11) into a vector problem that we know how to solve. Note that the spectral norm is unitary invariant since it is a Schatten norm, i.e., $\|A\| = \|UAV^\top\|$. So consider instead the SVD, $W = U \operatorname{diag}(\sigma) V^\top$, for which the dual norm simplifies,

$$\|W\| = \|U \operatorname{diag}(\sigma) V^\top\| = \|\sigma\|_\infty := \max_{\|x\|_1 \leq 1} \langle \sigma, x \rangle. \tag{12}$$

In other words, we need $\|\sigma\|_\infty = \langle \sigma, x \rangle$ which is attained by

$$x_j := \begin{cases} 1 & j \in \operatorname{Arg\,max}_{i \in [d]} \sigma_i \\ 0 & \text{otherwise} \end{cases} \quad \forall j \in [d]. \tag{13}$$

This solution also trivially satisfies the $\|\cdot\|_1$-constraint.

It remains to translate $x$ in (13) into a matrix solution $X$, which can be done by rewriting the objective and constraint in (12) back into the problem in (11),

$$\begin{aligned}
\|x\|_1 &= \|\operatorname{diag}(x)\|_* = \|U \operatorname{diag}(x) V^\top\|_* = \|\bar{X}\|_*, \\
\langle \sigma, x \rangle &= \operatorname{tr}(V^\top V \operatorname{diag}(\sigma) U^\top U \operatorname{diag}(x)) \\
&= \operatorname{tr}(V \operatorname{diag}(\sigma) U^\top U \operatorname{diag}(x) V^\top) = \langle U \operatorname{diag}(\sigma) V^\top, U \operatorname{diag}(x) V^\top \rangle = \langle W, U \operatorname{diag}(x) V^\top \rangle.
\end{aligned} \tag{14}$$

So it must be that $X = -\rho \bar{X} = -\rho U \operatorname{diag}(x) V^\top$. For our choice of $x$ the solution reduces to $X = -\rho u v^\top$. $\quad\square$

## B  Proofs for Section 4 (Convergence analysis)

**Theorem 4.3.** *Suppose Assumptions 4.1 and 4.2 hold with $L = \mu$. Then,* (10) *with the (arbitrary) norm perturbation $\varepsilon^k \in \operatorname{Arg\,max}_{\varepsilon: \|\varepsilon\|_* \leq 1} \langle \nabla f(x^k), \varepsilon \rangle$ satisfies the following descent inequality with $\rho \geq 0$ and $\gamma_k > 0$,*

$$f(x^{k+1}) - f^\star \leq (1 - \mu \gamma_k)(f(x^k) - f^\star) - \gamma_k(1 - \gamma_k L) L \rho \|\nabla f(x^k)\| + \frac{\gamma_k^2 L^3 \rho^2 \xi_*^2}{2}.$$

*Furthermore, for $\gamma_k = \gamma = \min\{\frac{1}{\mu K} \log(\alpha), \frac{1}{2L}\}$ with $\alpha = \frac{\Delta_0 2 \mu^3 K^2}{L^4 \rho^2 \xi_*^4 \xi^2}$, it follows that*

$$\min_{k \in [K]} f(x^k) - f^\star = \widetilde{\mathcal{O}}\big( \exp(-\tfrac{K}{2}) \Delta_0 + \tfrac{C \rho^2}{K^2} \big)$$

*with $\Delta_0 = f(x^0) - f^\star$, $C = L \xi_*^4 \xi^2$, $\xi = \max_{x \in \mathcal{X}} \|x\|_2 / \|x\|$ and $\xi_* = \max_{x \in \mathcal{X}} \|x\|_2 / \|x\|_*$.*

*Proof.* By using smoothness we have

$$\begin{aligned}
f(x^{k+1}) &\leq f(x^k) + \langle \nabla f(x^k), x^{k+1} - x^k \rangle + \frac{L}{2} \|x^{k+1} - x^k\|_2^2 \\
&= f(x^k) - \gamma_k \langle \nabla f(x^k), \nabla f(\widetilde{x}^k) \rangle + \frac{\gamma_k^2 L}{2} \|\nabla f(\widetilde{x}^k)\|_2^2 \\
&= f(x^k) - \gamma_k (1 - \frac{\gamma_k L}{2}) \|\nabla f(x^k)\|_2^2 + \frac{\gamma_k^2 L}{2} \|\nabla f(\widetilde{x}^k) - \nabla f(x^k)\|_2^2 \\
&\quad - \gamma_k (1 - \gamma_k L) \langle \nabla f(x^k), \nabla f(\widetilde{x}^k) - \nabla f(x^k) \rangle
\end{aligned} \tag{15}$$

Let us develop the last term of (15) with some $c > 0$:

$$
\begin{aligned}
\langle \nabla f(x^k), \nabla f(\widetilde{x}^k) - \nabla f(x^k) \rangle &= c \langle \rho \varepsilon^k, \nabla f(\widetilde{x}^k) - \nabla f(x^k) \rangle + \langle \nabla f(x^k) - c\rho\varepsilon^k, \nabla f(\widetilde{x}^k) - \nabla f(x^k) \rangle \\
&= c \langle \widetilde{x}^k - x^k, \nabla f(\widetilde{x}^k) - \nabla f(x^k) \rangle + \langle \nabla f(x^k) - c\rho\varepsilon^k, \nabla f(\widetilde{x}^k) - \nabla f(x^k) \rangle \\
\text{(Assumption 4.1)} &\geq \mu c \rho^2 \|\varepsilon^k\|_2^2 + \langle \nabla f(x^k) - c\rho\varepsilon^k, \nabla f(\widetilde{x}^k) - \nabla f(x^k) \rangle \\
&= \mu c \rho^2 \|\varepsilon^k\|_2^2 + c\rho\|\nabla f(x^k)\| - \|\nabla f(x^k)\|_2^2 + \langle \nabla f(x^k) - c\rho\varepsilon^k, \nabla f(\widetilde{x}^k) \rangle \\
&= (\mu - \tfrac{c}{2}) c \rho^2 \|\varepsilon^k\|_2^2 + c\rho\|\nabla f(x^k)\| - \tfrac{1}{2}\|\nabla f(x^k)\|_2^2 \\
&\quad - \tfrac{1}{2}\|\nabla f(x^k) - \nabla f(\widetilde{x}^k)\|_2^2 + \tfrac{1}{2}\|\nabla f(\widetilde{x}^k) - c\rho\varepsilon^k\|_2^2 \\
\text{(Assumption 4.2)} &\geq (\mu c - \tfrac{c^2}{2} - \tfrac{L^2}{2}) \rho^2 \|\varepsilon^k\|_2^2 + c\rho\|\nabla f(x^k)\| - \tfrac{1}{2}\|\nabla f(x^k)\|_2^2 + \tfrac{1}{2}\|\nabla f(\widetilde{x}^k) - c\rho\varepsilon^k\|_2^2 \\
&= (\tfrac{\mu}{c} - \tfrac{1}{2} - \tfrac{L^2}{2c^2})\|c\rho\varepsilon^k\|_2^2 + c\rho\|\nabla f(x^k)\| - \tfrac{1}{2}\|\nabla f(x^k)\|_2^2 + \tfrac{1}{2}\|\nabla f(\widetilde{x}^k) - c\rho\varepsilon^k\|_2^2
\end{aligned}
\tag{16}
$$

where we have used the definition of dual norm in the third equality. The first term of (16) is positive by assuming $\mu \geq \frac{c}{2} + \frac{L^2}{2c}$ which is feasible when $\mu = L$ (pick $c = L$).

$c = \sqrt{\mu/L}L$

Plugging (16) into (15) (and ignoring the last good terms of (16)) we get

$$
\begin{aligned}
f(x^{k+1}) &\leq f(x^k) - \tfrac{\gamma_k}{2}\|\nabla f(x^k)\|_2^2 - \gamma_k(1 - \gamma_k L) L\rho\|\nabla f(x^k)\| + \tfrac{\gamma_k^2 L}{2}\|\nabla f(\widetilde{x}^k) - \nabla f(x^k)\|_2^2 \\
\text{(Assumption 4.2)} &\leq f(x^k) - \tfrac{\gamma_k}{2}\|\nabla f(x^k)\|_2^2 - \gamma_k(1 - \gamma_k L) L\rho\|\nabla f(x^k)\| + \tfrac{\gamma_k^2 L^3 \rho^2}{2}\|\varepsilon^k\|_2^2 \\
&\leq f(x^k) - \tfrac{\gamma_k}{2}\|\nabla f(x^k)\|_2^2 - \gamma_k(1 - \gamma_k L) L\rho\|\nabla f(x^k)\| + \tfrac{\gamma_k^2 L^3 \rho^2 \xi_*^2}{2}.
\end{aligned}
$$

where the last line uses $\xi_* := \max_{x \in \mathcal{X}} \frac{\|x\|_2}{\|x\|_*}$ and $\|\nabla f(x^k)\|_* \leq 1$. From the PL condition, $f(x) - f^\star \leq \frac{\mu}{2}\|\nabla f(x)\|_2^2 \ \forall x \in \mathcal{X}$, (implied by strong convexity and Lipschitz continuity) it follows that

$$
f(x^{k+1}) - f^\star \leq (1 - \mu\gamma_k)(f(x^k) - f^\star) - \gamma_k(1 - \gamma_k L)L\rho\|\nabla f(x^k)\| + \tfrac{\gamma_k^2 L^3 \rho^2 \xi_*^2}{2}.
\tag{17}
$$

Set $\gamma_k = \gamma$ and consider two cases.

**Case I:** When $\|\nabla f(x^k)\| \geq \frac{\gamma L^2 \rho \xi_*^2}{2(1 - \gamma L)}$ for all $k$ up to $K$, the second last term $\gamma(1 - \gamma L)L\rho\|\nabla f(x^k)\|$ dominates, leading to geometric decay. We have

$$
f(x^K) - f^\star \leq (1 - \mu\gamma)^K (f(x^0) - f^\star).
$$

**Case II:** On the other hand, when $\|\nabla f(x^k)\| \leq \frac{\gamma L^2 \rho \xi_*^2}{2(1 - \gamma L)}$, we have from the PL condition that

$$
f(x^k) - f^\star \leq \|\nabla f(x^k)\|_2^2 \leq \tfrac{\xi^2}{2\mu}\|\nabla f(x^k)\|^2 \leq \frac{\gamma^2 L^4 \rho^2 \xi_*^4 \xi^2}{8\mu(1 - \gamma L)^2}.
$$

with $\xi := \max_{x \in \mathcal{X}} \frac{\|x\|_2}{\|x\|}$. At any $k$, either of the two cases hold, so we can upper bound by their sum as follows

$$
\begin{aligned}
\min_{k \in [K]} f(x^k) - f^\star &\leq (1 - \mu\gamma)^K \Delta_0 + \frac{\gamma^2 L^4 \rho^2 \xi_*^4 \xi^2}{8\mu(1 - \gamma L)^2} \\
&\leq (1 - \mu\gamma)^K \Delta_0 + \frac{\gamma^2 L^4 \rho^2 \xi_*^4 \xi^2}{2\mu} \\
&\leq \exp(-\mu\gamma K)\Delta_0 + \frac{\gamma^2 L^4 \rho^2 \xi_*^4 \xi^2}{2\mu},
\end{aligned}
$$

where for simplicity we have picked $\gamma \leq \frac{1}{2L}$ in the second last inequality and the last inequality uses $(1 - x)^K \leq \exp(-xK)$ for $x = \mu\gamma < 1$.

Consider the stepsize $\gamma = \min\{\frac{c}{\mu K}, \frac{1}{2L}\}$ where the scalar $c$ is to be decided. There are two cases.

First, if $\frac{1}{2L} > \frac{c}{\mu K}$ then $\gamma = \frac{c}{\mu K}$ and consequently

$$\min_{k \in [K]} f(x^k) - f^\star \le \exp(-c)\Delta_0 + \frac{c^2 L^4 \rho^2 \xi_*^4 \xi^2}{2\mu^3 K^2} \le (1+c^2)\frac{L^4 \rho^2 \xi_*^4 \xi^2}{2\mu^3 K^2}$$

where we have picked $c = \log(\frac{\Delta_0 2\mu^3 K^2}{L^4 \rho^2 \xi_*^4 \xi^2})$ to optimize the bound.

On the other hand, if $\frac{1}{2L} \le \frac{c}{\mu K}$ then $\gamma = \frac{1}{2L} \le \frac{c}{\mu K}$ and consequently

$$\min_{k \in [K]} f(x^k) - f^\star \le \exp(-\frac{\mu}{2L}K)\Delta_0 + \frac{c^2 L^4 \rho^2 \xi_*^4 \xi^2}{2\mu^3 K^2}$$

where the second term uses that $\gamma \le \frac{c}{\mu K}$.

In any case we can upper bound using the sum of the two bounds

$$\min_{k \in [K]} f(x^k) - f^\star \le \exp(-\frac{\mu}{2L}K)\Delta_0 + (2+c^2)\frac{L^4 \rho^2 \xi_*^4 \xi^2}{2\mu^3 K^2} = \widetilde{\mathcal{O}}(\exp(-\frac{\mu}{2L}K)\Delta_0 + \frac{L^4 \rho^2 \xi_*^4 \xi^2}{\mu^3 K^2}).$$

where $\widetilde{\mathcal{O}}$ hides logarithmic factors. This completes the proof.

$\square$

**Theorem 4.6.** *Suppose Assumptions 4.2 and 4.5 hold and $\|\varepsilon^k\| \le \xi$ for all $k \in [K]$. Then (10) satisfies the following convergence guarantee for $\rho_k \ge 0$ and $\gamma_k \in (0, 2/L)$,*

$$\min_{k=0,\dots,K-1} \|\nabla f(x^k)\|_2^2 \le \frac{\Delta_0 + C_1 \sum_{k=0}^{K-1} \gamma_k^2 \rho_k^2 + C_2 \sum_{k=0}^{K-1} \gamma_k \rho_k}{\sum_{k=0}^{K-1} \gamma_k(1-\gamma_k L/2)}$$

*with $\Delta_0 = f(x^0) - f^\star$, $C_1 = \frac{1}{2}L^3\xi^2$ and $C_2 = LG\xi$.*

*In particular, for constant stepsize $\gamma_k = \gamma = 1/L\sqrt{K}$ and $\rho = \frac{1}{\sqrt{K}}$, the following rate is obtained*

$$\min_{k=0,\dots,K-1} \|\nabla f(x^k)\|_2^2 = \mathcal{O}(\frac{\Delta_0}{\sqrt{K}} + \frac{C_2}{\sqrt{K}} + \frac{C_1}{LK^{3/2}})$$

*Proof.* Using smoothness we have

$$f(x^{k+1}) \le f(x^k) + \langle \nabla f(x^k), x^{k+1} - x^k \rangle + \frac{L}{2}\|x^{k+1} - x^k\|_2^2$$
$$= f(x^k) - \gamma_k \langle \nabla f(x^k), \nabla f(\widetilde{x}^k) \rangle + \frac{\gamma_k^2 L}{2}\|\nabla f(\widetilde{x}^k)\|_2^2$$
$$= f(x^k) - \gamma_k(1 - \frac{\gamma_k L}{2})\|\nabla f(x^k)\|_2^2 + \frac{\gamma_k^2 L}{2}\|\nabla f(\widetilde{x}^k) - \nabla f(x^k)\|_2^2$$
$$\quad - \gamma_k(1 - \gamma_k L)\langle \nabla f(x^k), \nabla f(\widetilde{x}^k) - \nabla f(x^k)\rangle$$
$$\text{(Assumption 4.2)} \le f(x^k) - \gamma_k(1 - \frac{\gamma_k L}{2})\|\nabla f(x^k)\|_2^2 + \frac{\gamma_k^2 \rho_k^2 L^3}{2}\|\varepsilon^k\|_2^2 - \gamma_k(1 - \gamma_k L)\langle \nabla f(x^k), \nabla f(\widetilde{x}^k) - \nabla f(x^k)\rangle.$$
$$(18)$$

What remains is the last term of (18):

$$-\langle \nabla f(x^k), \nabla f(\widetilde{x}^k) - \nabla f(x^k)\rangle \le \|\nabla f(x^k)\|_2 \|\nabla f(\widetilde{x}^k) - \nabla f(x^k)\|_2$$
$$\text{(Assumption 4.2)} \le \rho_k L \|\varepsilon^k\|_2 \|\nabla f(x^k)\|_2$$
$$\text{(Assumption 4.5)} \le \rho_k LG \|\varepsilon^k\|_2$$
$$(19)$$

Plugging (19) back into (18) and using that $\|\varepsilon^k\|_2 \le \xi$ we have

$$f(x^{k+1}) \le f(x^k) - \gamma_k(1 - \frac{\gamma_k L}{2})\|\nabla f(x^k)\|_2^2 + \frac{\gamma_k^2 \rho_k^2 L^3 \xi^2}{2} + \gamma_k(1 - \gamma_k L)\rho_k LG\xi.$$

Subtracting $f^\star$ on both sides, telescoping and rearranging completes the proof. $\square$

## C  Implementation

---

**Algorithm 3** Pseudo-code for distributed implementation of $\nu$SAM.

---

1: //Compute the gradient on each device over a distinct batch:
2: grad ← compute_gradients(model, batch)
3: **if** not m-SAM **then**
4:     //Average and synchronize the gradients:
5:     grad ← all_reduce(grad, operation = "*mean*")
6: **end if**
7: //Perturb the model (while maintaining a sparse representation of the perturbation):
8: sparse_eps ← compute_perturbation(model, grad)
9: model ← add_sparse_perturbation(model, sparse_eps)
10: //Compute the gradient on the perturbed model on each device and synchronize:
11: grad ← compute_gradients(model, batch)
12: grad ← all_reduce(grad, operation = "*mean*")
13: //Remove the (possibly distinct) perturbation on each device:
14: model ← subtract_sparse_perturbation(model, sparse_eps)
15: //At this point model and grad are identical across all devices:
16: model ← optimizer_step(model, grad)

---

### C.1  $\nu$SAM variant with normalization

In this section, we derive for completeness a variant of $\nu$SAM which normalizes the layerwise perturbations, that we refer to as $\nu$SAM-Norm.

It follows from the Eckart-Young-Mirsky theorem that a rank-1 approximation to a matrix $A \in \mathbb{R}^{n \times m}$ is given by its top singular value $\sigma$ and associated singular vectors $u, v$ as follows

$$\underset{X:\mathrm{rank}(X) \leq 1}{\arg\min} \|A - X\|_2 = \sigma u v^\top. \tag{20}$$

Suppose for simplicity that the network parameters are composed only of matrices, i.e., $x = (M_1, ..., M_l)$ such that $\nabla f(x) = (\nabla_{M_1} f(x), ..., \nabla_{M_l} f(x))$. Define $\sigma_i$ to be the top singular value and $u_i, v_i$ the associated singular vectors of $\nabla_{M_i} f(x)$. Then the SAM perturbation update on the rank-1 approximation instead of the gradient reduces to the following

$$\varepsilon_i = \rho \sigma_i u_i v_i^\top / c \quad \text{with} \quad c = \|(\sigma_1 u_1 v_1^\top, ..., \sigma_l u_l v_l^\top)\|_2 = \sqrt{\textstyle\sum_j \sigma_j^2}. \tag{21}$$

In other words, the resulting ascent update for $\nu$SAM-Norm is identical to Step 2 in $\nu$SAM but with a scaling factor of $\sigma_i / \sqrt{\sum_{j=1}^{l} \sigma_j^2}$ for the $i^{\text{th}}$ matrix. Similar to $\nu$SAM, the perturbation update can be seen as a norm-constrained LMO on $x$, so convergence gaurantees of Theorem 4.3 also applies to $\nu$SAM-Norm. Experimental results can be found in Table 8 of Appendix D, where we find that $\nu$SAM-Norm has improved performance in certain cases, but is generally dominated by $\nu$SAM.

## D  Experiments

- For training from scratch see Tables 6 to 10 and Figure 4.

- For fine-tuning see Tables 11 to 13 and Figure 5.

- Computational resources and wall-clock time are specified in Appendix D.1.

Table 6: The architectures used for CIFAR10/100 training from scratch. We take the original model configurations used in Dosovitskiy et al. (2020); Tolstikhin et al. (2021) but changes the patch size to 4 to accommodate for the smaller image size of $32 \times 32$.

| Model | Embedding dim. | Depth | Heads | Patch size |
|-------|----------------|-------|-------|------------|
| ViT-S/4 | 384 | 12 | 6 | 4 |
| ViT-B/4 | 768 | 12 | 12 | 4 |
| Mixer-S/4 | 512 | 8 | - | 4 |
| Mixer-B/4 | 768 | 12 | - | 4 |

Table 7: Baseline (AdamW) hyperparameters for training from scratch. We use a cosine learning rate schedule with linear warmup. We use standard augmentations (random cropping and flipping) and AutoAugment.

| Hyperparameter | Value |
|----------------|-------|
| Learning rate | 0.0005 |
| Label smoothing | 0.1 |
| Weight decay | 0.05 |
| Warmup epoch | 10% |
| Epochs | 300 |
| Dropout rate | 0.0 |
| Drop path rate | 0.1 |
| Gradient clipping | Disabled |
| Batch size | 128 |

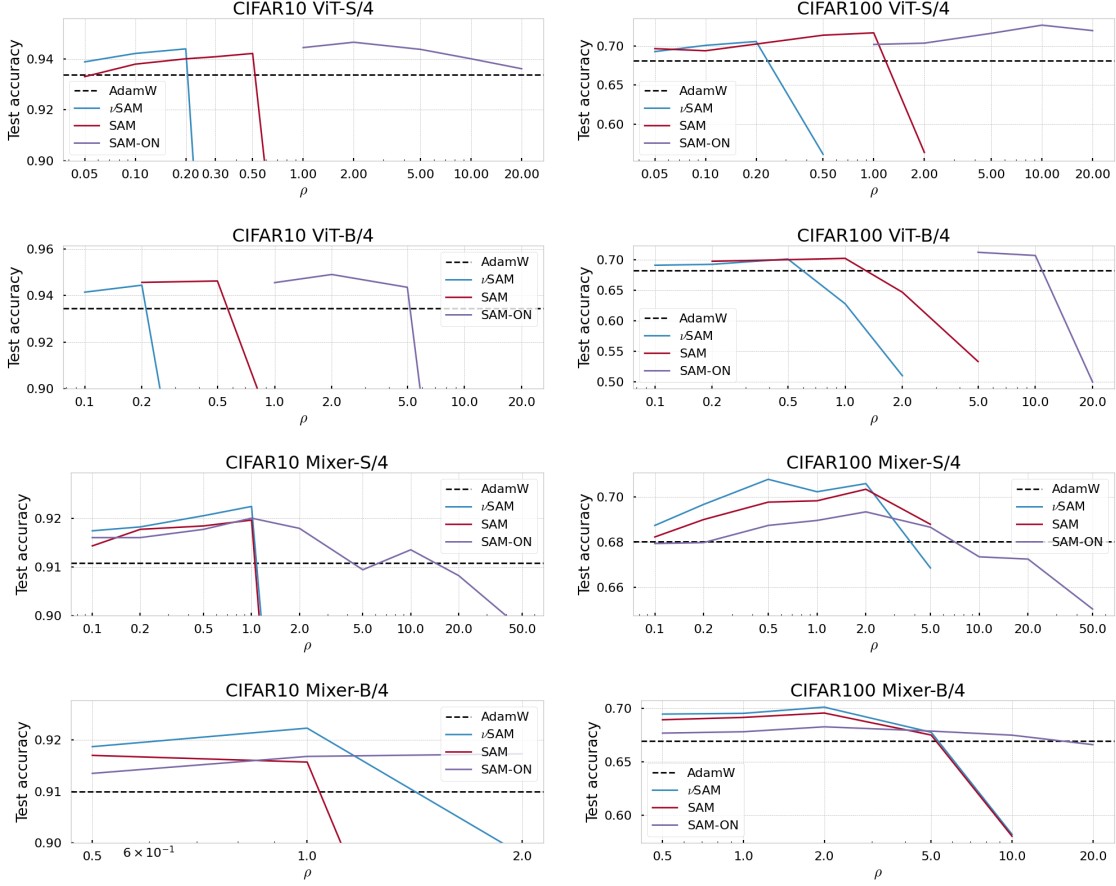

Figure 4: Sweep over $\rho$ for pretraining results in Table 2.

Table 8: $\nu$SAM-Norm variant on ViT-S/4 and Mixer-S/4 (see Appendix C.1 for the formulation).

| Dataset | Method
Perturbation start epoch
Model | AdamW
0 | $\nu$SAM
30 | $\nu$SAM-Norm
30 |
|---|---|---|---|---|
| CIFAR10 | Mixer-S/4 | 91.08 | **92.24** (+1.16) | 92.10 (+1.02) |
| | ViT-S/4 | 93.37 | **94.39** (+1.02) | 93.95 (+0.58) |
| CIFAR100 | Mixer-S/4 | 68.03 | **70.79** (+2.76) | 70.17 (+2.14) |
| | ViT-S/4 | 68.17 | 70.60 (+2.43) | **71.13** (+2.96) |

Table 9: SAM-ON with perturbation delay interestingly does not improve the test accuracy.

| Dataset | Perturbation start epoch
Model | 0 | 30 |
|---|---|---|---|
| CIFAR100 | ViT-S/4 | **72.70** | 71.85 |

Table 10: Ablation of the number of power iterations. Maybe surprisingly, increasing the number of power iterations does not necessarily lead to higher test accuracies. One possible explanation is that the warmstart in the power iterations (Algorithm 2) might act as a variance reduction like mechanism by implicitly incorporating previous stochastic gradients in the approximation. Applying variance reduction to the perturbation direction has shown to empirically be beneficial (Li & Giannakis, 2024). Investigating this connection is interesting future work.

| Model | Method
Power iterations
Dataset | $\nu$SAM
1 | 50 |
|---|---|---|---|
| ViT-B/4 | CIFAR10 | 94.44 | 94.35 |
| | CIFAR100 | 70.15 | 70.27 |
| ViT-S/4 | CIFAR10 | 94.39 | 94.47 |
| | CIFAR100 | 70.60 | 70.24 |

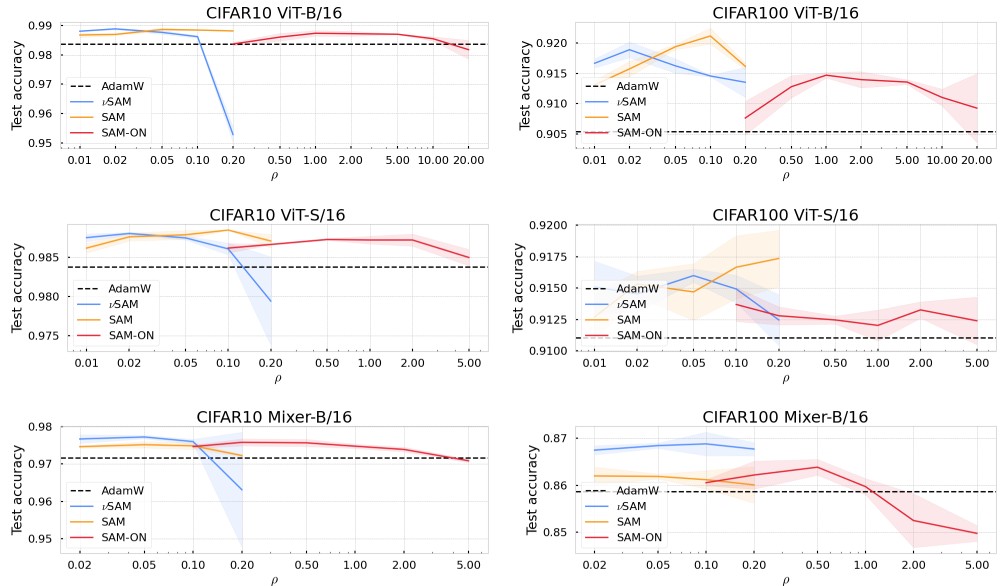

Figure 5: Sweep over $\rho$ for fine-tuning results in Table 4. Mean and standard deviation for each configuration is computed over 3 independent runs.

Table 11: Baseline (AdamW) hyperparameters for fine-tuning. We use a cosine learning rate schedule.

| Hyperparameter | Value |
| --- | --- |
| Learning rate | 0.0001 |
| Label smoothing | 0.0 |
| Weight decay | 0.0 |
| Warmup epoch | 0 |
| Epochs | 10 |
| Dropout rate | 0.0 |
| Drop path rate | 0.0 |
| Gradient clipping | Disabled |
| Batch size | 96 |

Table 12: Hyperparameters for fine-tuning on GLUE.

| Hyperparameter | Adam | SAM[1] | $\nu$SAM [1] | SAM-ON[1] |
| --- | --- | --- | --- | --- |
| Model | Pretrained BERT base (uncased) (Devlin et al., 2018) | | | |
| Epochs | 5 | | | |
| Batch size | 32 | | | |
| Learning rate | $4 \cdot 10^{-5}$ (cosine schedule) | | | |
| Learning rate warmup | 10% | | | |
| $\rho$ | - | { $0.01, \mathbf{0.02}, 0.05$ } | { $\mathbf{0.01}, 0.02, 0.05$ } | { $0.01, 0.02, \mathbf{0.05}$ } |
| Perturbation delay | - | 10% | 10% | 0% |

[1] All methods uses Adam as the base optimizer.

Table 13: Even if AdamW is given a more refined budget to optimize the learning rate ($lr = 5e−05$), ($\nu$)SAM with the same learning rate improves even further when using the default parameter $\rho = 0.05$ ($\rho = 0.02$). Providing AdamW with twice the computational budget also does not close the gap.

|  |  | Test accuracy |
|---|---|---|
| lr=0.0005 | AdamW | **97.19** |
| lr=0.0001 | AdamW (2x epochs) | 98.49 (+0.11) |
|  | AdamW | 98.38 |
|  | SAM | **98.76** (+0.38) |
| lr=0.00005 | AdamW (2x epochs) | 98.49 (-0.13) |
|  | AdamW | 98.62 |
|  | SAM | **98.89** (+0.27) |
|  | $\nu$SAM | 98.81 (+0.19) |
| lr=0.00001 | AdamW | 98.44 |
|  | SAM | **98.72** (+0.28) |

## D.1 Computational resources

All experiments are run on either a single NVIDIA V100 GPU. Table 14 shows the wall-clock time.

Table 14: Wall-clock time for training ViTs and MLP-Mixer models from scratch on a single V100 GPU. Both $\nu$SAM (with one power iteration) and SAM require the same computational complexity.

| Dataset | Model | Method | minutes / epoch |
|---|---|---|---|
| CIFAR10 | Mixer-B/4 | $\nu$SAM | 4.4 |
|  |  | SAM | 4.4 |
|  | Mixer-S/4 | $\nu$SAM | 1.6 |
|  |  | SAM | 1.6 |
|  | ViT-B/4 | $\nu$SAM | 4.3 |
|  |  | SAM | 4.2 |
|  | ViT-S/4 | $\nu$SAM | 1.6 |
|  |  | SAM | 1.7 |
| CIFAR100 | Mixer-B/4 | $\nu$SAM | 4.4 |
|  |  | SAM | 4.3 |
|  | Mixer-S/4 | $\nu$SAM | 1.6 |
|  |  | SAM | 1.6 |
|  | ViT-B/4 | $\nu$SAM | 4.3 |
|  |  | SAM | 4.2 |
|  | ViT-S/4 | $\nu$SAM | 1.7 |
|  |  | SAM | 1.7 |

