# OpenReview forum: "νSAM: Memory-Efficient Sharpness-Aware Minimization via Nuclear Norm Constraints"
_TMLR — Accepted by TMLR_

### Review · Reviewer_ffMj · 2024-11-07

**Summary Of Contributions:**

This paper proposes $\nu$SAM, a method designed to address the increased memory footprint associated with SAM training. $\nu$SAM employs a layer-wise low-rank perturbation that leads to substantial memory savings. The paper provides convergence analysis and extensive experiments on various datasets.

**Audience:**

Yes

**Broader Impact Concerns:**

No ethical concerns

**Claims And Evidence:**

Yes

**Requested Changes:**

1. On Page 4, it would be helpful to provide references for the statement: "since including the parameters did not improve performance and could even sometimes lead to performance degradation."
2. Include experiments on more commonly used architectures, such as ResNet, VGG, and Wide-ResNet, particularly for CIFAR-10/100.
3. It would be beneficial to provide convergence analysis without relying on the strongly-convex assumption, which is rarely valid in DNNs.
4. Additionally, many other works have applied random weight perturbation to improve generalization and have shown promising performance (e.g., [1], [2]). It would be helpful to discuss these works when claiming that "normalized Gaussian noise (GaussSAM) exhibits non-convergence."

References:

[1] Bisla et al. "Low-pass filtering SGD for recovering flat optima in the deep learning optimization landscape." AISTATS 2022.

[2] Li et al. "Revisiting Random Weight Perturbation for Efficiently Improving Generalization." TMLR 2024.

**Strengths And Weaknesses:**

**Strengths:**
1. The paper tackles an important problem on the additional memory burden of SAM, particularly relevant when fine-tuning large models such as large language models (LLMs).
2. The writing is clear, and the paper is well-structured and easy to follow.
3. The experiments are thorough and provide convincing evidence for the effectiveness of the approach.
4. The proposed method is both sound and reasonable. Interpreting SAM's perturbation from a norm perspective is particularly interesting.

**Weaknesses:**
1. $\nu$SAM does not demonstrate a significant advantage over SAM-ON, which is the primary limitation of this approach. Note that SAM-ON also does not impose a significant memory burden compared to SGD and is much simpler.
2. The assumptions in the convergence analysis, such as Assumption 4.1, are quite strong. For example, the strongly-convex assumption is rarely satisfied in practice, particularly for deep neural networks (DNNs).
3. The paper lacks experiments with more commonly used architectures, such as ResNet, VGG, and Wide-ResNet.
4. It appears that the current approach can only be applied to matrix weights in DNNs. A broader explanation of how the method could handle matrices with more than one dimensionality would be beneficial.

---

> ### Author Response · Authors · 2024-11-26
>
> We thank the reviewer for their feedback and address the remaining concerns below.
>
> > "$\nu$SAM does not demonstrate a significant advantage over SAM-ON, which is the primary limitation of this approach."
>
> We agree that SAM-ON is a very useful memory-efficient method, which is why we have provided extensive comparison against the method.
>
> The main reasons for providing an alternative in the form of $\nu$SAM is that $\nu$SAM does provide an advantage over SAM-ON in some settings. E.g. i) $\nu$SAM _consistently_ improves over SAM-ON for fine-tuning on image tasks (Table 4) ii) $\nu$SAM provides consistent improvement when training MLP-Mixer models from scratch (Table 2-3).
>
> Additionally, $\nu$SAM perturbs all weight matrices whereas SAM-ON only perturbs layer norm weights. This could have potential drawbacks, and we would argue that it is beneficial to have an alternative memory-efficient method like $\nu$SAM available that perturbs all weight matrices.
>
> > Strong assumptions of Theorem 4.3
>
> Theorem 4.6 provides convergence for non-convex problems.
> However, for fixed perturbation this seems to be much more challenging.
> We have added a comment mentioning that assumptions in Theorem 4.3 are strong both after the theorem and under contributions.
>
> > ResNet, VGG, and Wide-ResNet
>
> The $\nu$SAM method indeed requires matrix structure, which is why we focus on transformer and MLP-Mixer architectures.
> We have included a paragraph explicitly stating this limitation right after stating the contributions.
>
> > "On Page 4, it would be helpful to provide references for the statement: 'since including the parameters did not improve performance and could even sometimes lead to performance degradation.'"
>
> This was an observation made in the early experimental phase, where we tried both perturbing biases with $\|\cdot\|_\infty$ and $\|\cdot\|_2$ constraints on fine-tuning specifically. We have removed the statement to avoid potentially too strong a claim.
>
> > Non-convergence of random weight perturbation
>
> Our comment regarding non-convergence is not in conflict with the existing literature, but rather supported by it.
> For instance, Theorem 4 in [2] only shows convergences to a $\sigma^2$ neighborhood of the solution instead of the solution itself ($\sigma$ being the perturbation noise level).
> We have expanded the discussion to include the reference to theory.

---

### Review · Reviewer_Utmq · 2024-11-15

**Summary Of Contributions:**

This paper proposes a variant of sharpness-aware minimization (SAM) based on nuclear norm constraints on the ascent step, which uses 1/3 memory compared to the vanilla SAM, while achieving comparable performance.

**Audience:**

Yes

**Broader Impact Concerns:**

Not applicable.

**Claims And Evidence:**

No

**Requested Changes:**

I find Theorem 4.3 to be too restrictive. Even without assuming $f$ is quadratic, having $\mu=L$ as an assumption is not convincing. Experimental results are also somewhat inconclusive—it’s hard for me to find a reason to use $\nu$SAM compared to SAM-ON, for example. I think addressing these points are important.

**Strengths And Weaknesses:**

***Strength***

The paper is generally well-written. Convergence theory is provided, although seems quite restrictive. Experimental results including training from scratch, as well as fine-tuning for language and vision tasks are included.


***Weaknesses/questions***
- Why is SAM framed as injecting noise to the gradient?… Claim about perturbing with Gaussian noise and not leading to convergence is quite misleading. SAM does NOT do that; it takes a gradient ascent step.
- Nuclear norm constraint is defined as $\sum_i \sigma_i(A)$. How does Eq (7a) and (7b) follow? It just adds a rank-1 matrix.
- I’m also a bit confused about the claim of Theorem 4.3. First, where do I see that $f$ is quadratic? Second, why is it reasonable to assume $L = \mu$? Third, there already exists much more general theory: Theorem 2 in [2]. I find it a bit misleading to compare with the generalization bound from Foret et al (2020, Thm1).
- If $f$ is strongly convex quadratic with $\mu = L$, then the Hessian has uniform eigenvalues, or the hessian has condition number 1. Am I missing something? This is an extremely restrictive setting.
- Delaying the perturbation also has been considered in [3], with theoretically principled explanation.
- Experimental results are not very convincing. SAM-ON, which has similar compression effect as the proposed method, works better in quite many settings.
- I believe the practical concern about SAM is having to evaluate gradient twice, more than the storage issue. In this regard, how does this work compare with [1]?

[1] J. Du et al. (2022) “Sharpness-Aware Training for Free”

[2] M. Andriushchenko et al (2022) “Towards Understanding Sharpness-Aware Minimization”

[3] A. Agarwala “SAM operates far from home”

---

> ### Author Response · Authors · 2024-11-26
>
> We thank the reviewer for their feedback and address the remaining concerns below.
>
> > "Why is SAM framed as injecting noise to the gradient?"
>
> We believe there is a misunderstanding.
> We do _not_ claim that SAM perturbs the weights with Gaussian noise.
> Section 4 and Figure 1, which considers random (Gaussian) perturbations, includes both SAM (which uses gradients for the perturbation) and what we refer to as GaussSAM (which uses random perturbations).
> The comment regarding non-convergence in Figure 1 applies to GaussSAM with fixed perturbation size, whereas we observe that both SAM and $\nu$SAM indeed converges.
>
> > "Nuclear norm constraint is defined as $\sum_i \sigma_i(A)$. How does Eq (7a) and (7b) follow? It just adds a rank-1 matrix."
>
> The update indeed simply adds a rank-1 update, which is the closed form expression to the LMO (see Section 3.1 referenced after Eq. 7a-b and the associated Appendix A for proof).
>
> > $f$ being quadratic and restrictiveness of assumptions
>
> The objective $f$ is quadratic due to $L=\mu$. We have now commented explicitly on this in the writeup immediately after the theorem.
> We agree that these conditions are strong, which is why we mention that Theorem 4.3 only applies to certain convex quadratics (we now emphasize it further under contributions).
> Although restrictive, this setting is sufficient for demonstrating the non-convergence behaviour of random perturbations (Figure 1).
> For the nonconvex case we provide Theorem 4.6.
>
> > "there already exists much more general theory: Theorem 2 in [2]."
>
> Theorem 2 of [2] is different in the following ways:
>
> - The theorem considers a non-normalized version of SAM, so the perturbation automatically becomes smaller as the iterates approach the solution.
> - The theorem requires a tiny perturbation radius (horizon dependent)
> - The theorem does not apply to arbitrary norm constraints
>
> > "I find it a bit misleading to compare with the generalization bound from Foret et al (2020, Thm1)."
>
> We believe there is a misunderstanding.
> We are not comparing our theorem with generalization bounds.
> Rather we are making the argument that a reasonable criterion for our method should be that it can at least minimize the original objective $\min_x f(x)$. This is the case for e.g. SAM, but not for GaussSAM as illustrated.
> Theorem 4.3 shows that $\nu$SAM has this same property as SAM.
>
> > "Delaying the perturbation also has been considered in [3], with theoretically principled explanation."
>
> We thank the reviewer for the reference, and have cited the work.
>
> The observation in [3] is somewhat different.
> In [3] they state that "Our analysis of the quadratic regression model suggests that SAM already regularizes the large eigenmodes at early times". Our empirical observation is rather that it improve performance to disable the perturbation during the warmup phase, when training ViT and MLP-Mixer models.
>
> > "Experimental results are not very convincing. SAM-ON, which has similar compression effect as the proposed method, works better in quite many settings."
>
> We agree that SAM-ON is a very useful memory-efficient method, which is why we have provided extensive comparison against the method.
>
> The main reasons for providing an alternative in the form of $\nu$SAM is that $\nu$SAM does provide an advantage over SAM-ON in some settings. E.g. i) $\nu$SAM _consistently_ improves over SAM-ON for fine-tuning on image tasks (Table 4) ii) $\nu$SAM provides consistent improvement when training MLP-Mixer models from scratch (Table 2-3).
>
> Additionally, $\nu$SAM perturbs all weight matrices whereas SAM-ON only perturbs layer norm weights. This could have potential drawbacks, and we would argue that it is beneficial to have an alternative memory-efficient method like $\nu$SAM available that perturbs all weight matrices.
>
> > "I believe the practical concern about SAM is having to evaluate gradient twice, more than the storage issue. In this regard, how does this work compare with [1]?"
>
> We agree that reducing the computational time of SAM is also interesting, but this is orthogonal to our work.

---

### Review · Reviewer_WxYQ · 2024-11-18

**Summary Of Contributions:**

This work proposes νSAM, a variant of Sharpness-aware Minimization that significantly reduces memory requirements by modifying the perturbation constraint. This achieves nearly 1/3 of the memory footprint compared to standard SAM while maintaining comparable performance. They demonstrate the effectiveness of νSAM through comprehensive experiments. They provide theoretical foundations by proving convergence guarantees for SAM with arbitrary norm choice, even with fixed perturbation radius. This extends the theoretical understanding of SAM-based optimization.

**Audience:**

Yes

**Claims And Evidence:**

Yes

**Requested Changes:**

* A formal analysis of the method's time complexity compared to standard SAM would strengthen the paper
* It's better to use some commonly-used notations

**Strengths And Weaknesses:**

Strengths:
* The paper demonstrates excellent writing quality and organization, making complex concepts accessible and maintaining a clear logical flow throughout. The presentation of ideas is systematic and well-structured.
* The experimental validation is comprehensive.
* The theoretical foundations are rigorously developed, with formal proofs for convergence guarantees under arbitrary norm choice.

Weaknesses:
* The notation choices deviate from common conventions in optimization literature, particularly: Using $x$ to denote model weights instead of the more standard $w$ or $\theta$
* Computational efficiency trade-offs need clearer discussion: While the method achieves significant memory savings, the time complexity implications from the additional SVD computation are not thoroughly analyzed.

---

> ### Author Response · Authors · 2024-11-26
>
> We thank the reviewer for their feedback and address the remaining concerns below.
>
> > Notational choice
>
> Using $x$ for the decision variable is fairly standard in the optimization for machine learning community.
> We prefer sticking to this notation, but will consider including a more explicit description of the notation if this could be helpful.
>
> > Time complexity discussion
>
> Please see Appendix D.1, which shows that $\nu$SAM has the same computation time as SAM (when power iteration = 1 as used in practice).

---

> > ### Comment · Reviewer_WxYQ · 2024-12-03
> > **Thank the authors**
> >
> > I thank the authors for their responses. I am sorry for that I cannot provide a thorough and competent review as I do not possess sufficient knowledge, but I think this manuscript is interesting for general readers.

---

### Review · Reviewer_MzVp · 2024-11-19

**Summary Of Contributions:**

The paper proposes low rank perturbations instead of the inf norm based pertubations done previously for SAM. this results in substantial savings on memory.

**Audience:**

Yes

**Claims And Evidence:**

Yes

**Requested Changes:**

- can u elaborate on table 1. how are there 22M params stored for perturbation instead of only 75k. i understand per iteration costs as explained on page4,5 but those still dont account for such a large discrepancy.

- some of the competitors need more details within the paper when discussing the related sections. especially sam-on.

- some of the writing needs cleaned up. please expand on how the nuclear norm constraint as claimed in the front matter of the paper is implied when using rank-one updates.

- have u tried combining sam-on and vsam? Why not just do low rank perturbations on the normalization layers?

**Strengths And Weaknesses:**

Strengths:

- The paper is well-written, with little need for improvement in writing.
- the empirical results are sound and adequate.
- some of the additional insights such as delay in perturbation are interesting.

Weaknesses
- there is almost no new theoretical results. the presented theorems are a rehash of previously known theorems applied to the low rank setting.
- given the sam-on paper, the results are mixed at best. i am not sure if the method itself is very useful unless further combined, investigated vs sam-on.

---

> ### Author Response · Authors · 2024-11-26
>
> We thank the reviewer for their feedback and address the remaining concerns below.
>
> > On SAM-ON
>
> We agree that SAM-ON is a very useful memory-efficient method, which is why we have provided extensive comparison against the method.
>
> The main reasons for providing an alternative in the form of $\nu$SAM is that $\nu$SAM does provide an advantage over SAM-ON in some settings. E.g. i) $\nu$SAM _consistently_ improves over SAM-ON for fine-tuning on image tasks (Table 4) ii) $\nu$SAM provides consistent improvement when training MLP-Mixer models from scratch (Table 2-3).
>
> Additionally, $\nu$SAM perturbs all weight matrices whereas SAM-ON only perturbs layer norm weights. This could have potential drawbacks, and we would argue that it is beneficial to have an alternative memory-efficient method like $\nu$SAM available that perturbs all weight matrices.
>
> > "How are there 22M params stored for perturbation instead of only 75k"
>
> For a given matrix $\nu$SAM only have to store 2 vectors (instead of the matrix). So instead of storing $n \times m$ matrix we only need to store $n+m$.
> We comment on this memory saving e.g. after the update in Eq. 8.
>
> > Expand related work on SAM-ON
>
> We are now explicitly mentioning the strong empirical performance and simplicity of SAM-ON.
>
> > How nuclear norm is implied when using rank-one update
>
> Section 3 (and specifically Section 3.1) is dedicated to explaining how the nuclear norm constraint and the rank-one update is connected.
> Specifically, the low rank update is obtained as the closed form solution to the LMO with nuclear norm constraints as provided in Lemma 3.1.
>
> > Combining with SAM-ON
>
> It is indeed possible to combine SAM-ON and $\nu$SAM (in this sense the methods are orthogonal).
> However, since the memory requirement is already negligible with either methods we do not consider this.

---

### Decision · Action_Editor_EgDo · 2024-12-30

**Recommendation:** Accept as is

**Comment:**

The paper studies an important topic of optimization with the goal of keeping the model generalization at a good level. While the proposed approach has a clear limitation due to its design based on the matrix structure, as implied by the low-rank approach, it shows promising results on practical benchmarks and allows for significant memory savings. The paper is very relevant to TMLR and its claims are supported by sufficient empirical evidence, which is why I believe it should be accepted.

**Audience:**

The reviewers noted that some of the results are going to be of interest to TMLR readers. The reviewers all agreed on the audience criterion and it is noted by several of them that the paper is written well and presents complicated concepts in an accessible way.

**Claims And Evidence:**

The reviewers were unanimous in confirming the work made claims that are supported by evidence. While they noted that the theoretical results have limited novelty and are a bit restrictive, the theory is rigorous and provides convergence guarantees for the method. The emprical result have been critized for being limited and not including networks such as ResNets, but the authors admitted it is a limitation of their approach, which requires matrix structure and they clearly stated that in the text of their paper.